# Beyond Fixed:
# Training-Free Variable-Length Denoising for Diffusion Large Language Models

**Jinsong Li**[1,2*] **Xiaoyi Dong**[1,2†] **Yuhang Zang**[2] **Yuhang Cao**[2] **Jiaqi Wang**[2,4†] **Dahua Lin**[1,3]

[1] The Chinese University of Hong Kong    [2] Shanghai AI Laboratory
[3] CPII under InnoHK    [4] Shanghai Innovation Institute
lj024@ie.cuhk.edu.hk
Github: https://github.com/Li-Jinsong/DAEDAL

## ABSTRACT

Diffusion Large Language Models (DLLMs) are emerging as a powerful alternative to the dominant Autoregressive Large Language Models, offering efficient parallel generation and capable global context modeling. However, the practical application of DLLMs is hindered by a critical architectural constraint: the need for a statically predefined generation length. This static length allocation leads to a problematic trade-off: insufficient lengths cripple performance on complex tasks, while excessive lengths incur significant computational overhead and sometimes result in performance degradation. While the inference framework is rigid, we observe that the model itself possesses internal signals that correlate with the optimal response length for a given task. To bridge this gap, we leverage these latent signals and introduce **DAEDAL**, a novel training-free denoising strategy that enables **D**ynamic **A**daptive Length **E**xpansion for **D**iffusion L**a**rge **L**anguage Models. DAEDAL operates in two phases: 1) Before the denoising process, DAEDAL starts from a short initial length and iteratively expands it to a coarse task-appropriate length, guided by a sequence completion metric. 2) During the denoising process, DAEDAL dynamically intervenes by pinpointing and expanding insufficient generation regions through mask token insertion, ensuring the final output is fully developed. Extensive experiments on DLLMs demonstrate that DAEDAL achieves performance comparable, and in some cases superior, to meticulously tuned fixed-length baselines, while simultaneously enhancing computational efficiency by achieving a higher effective token ratio. By resolving the static length constraint, DAEDAL unlocks new potential for DLLMs, bridging a critical gap with their Autoregressive counterparts and paving the way for more efficient and capable generation.

## 1 INTRODUCTION

Diffusion Large Language Models (DLLMs) (Nie et al., 2025; DeepMind, 2025; Inception Labs et al., 2025) have recently emerged as a promising paradigm for Large Language Models (LLMs) (Yang et al., 2025; Achiam et al., 2023), garnering significant attention from both academia and industry. Unlike the conventional autoregressive (AR) framework, which generates text sequentially via next-token prediction, DLLMs operate through a multi-step iterative denoising process. By leveraging bidirectional attention to refine an initially masked sequence into a coherent output, it offers several distinct advantages (Nie et al., 2024), including the ability to utilize global context for tasks that require holistic planning, a flexible trade-off between the number of inference steps and the quality of the generated sample. Given these unique properties, DLLMs have become a compelling research direction that offers an alternative to the autoregressive paradigm.

Despite their promising potential, the inference of DLLMs suffers from a fundamental limitation rooted in their denoising paradigm: the denoise starts from a fully-masked sequence of a fixed length,

---

*This work is done during an internship in Shanghai AI Laboratory. † Corresponding author

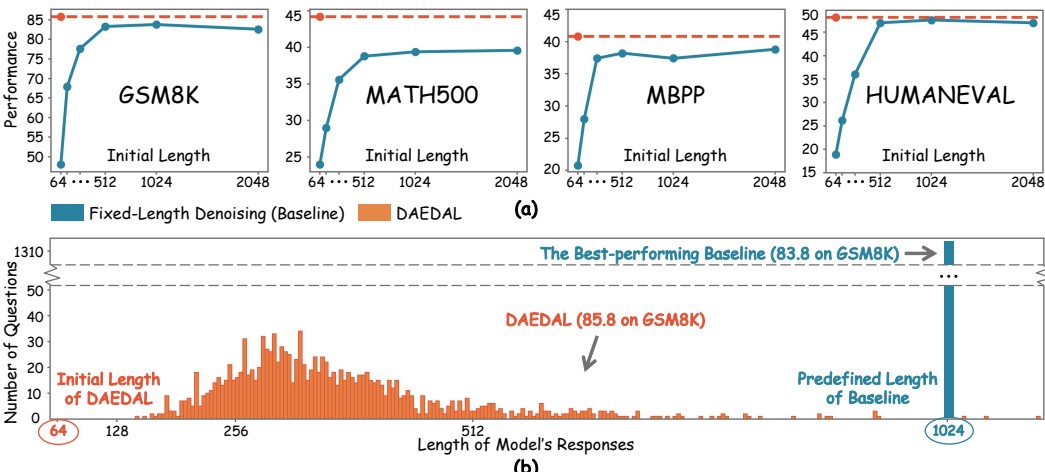

Figure 1: **Overview of DAEDAL's effectiveness on LLaDA-Instruct-8B. (a)** DAEDAL uses a unified and short initial length, consistently surpassing the baseline, which needs its length meticulously tuned for each benchmark to achieve peak performance. **(b)** DAEDAL dynamically adjusts length and adaptively expands on a per-problem basis, resulting in a varied distribution of response lengths. In contrast, the baseline is constrained to a fixed length for all problems.

hence the final output length is also statically predefined. This static setup leads to flawed inference compared to their autoregressive counterpart: AR LLMs flexibly adjust the output length based on the given task, while DLLMs must adapt the task output based on the given length.

The requirement of a manually pre-defined length leads to a severe dilemma. An overly short length hinders the model from solving complex problems due to an insufficient token length. Conversely, universally adopting a long generation length introduces a new set of issues. First, it incurs large computational overhead due to the quadratic complexity of bidirectional attention. Second, as shown in Figure 1 (a), we observe that excessively long initial lengths may degrade model performance.

This rigid, pre-defined length constraint not only creates the dilemma at the outset of inference, but more fundamentally, it cripples the model's ability to adapt dynamically. For instance, DLLMs lack the critical test-time scaling capability (Muennighoff et al., 2025) of AR models, which can extend their output to self-correct (e.g., "Wait, let me rethink...") (Shah et al., 2025; Wei et al., 2022). This problem is exacerbated by the non-sequential generation nature of DLLMs. A model might generate the beginning and end of a sequence first, only to find the allocated space for intermediate reasoning insufficient, leading directly to incomplete logic and degraded performance.

Fortunately, we find the solution lies within the DLLM's intrinsic capabilities — its planning ability. In each denoise step, the model plans the final output by predicting all the mask tokens with different confidence. We discovered that the model's prediction confidence acts as a powerful, general-purpose signal indicating whether the generation space is sufficient. For example, as shown in Figure 2, in the first denoising step (t=1), the model confidently predicts more End-of-Sequence tokens (EOS) from the fully masked sequence when the length is sufficient for the given task, while less confident in predicting EOS when the length is insufficient.

This core insight paves the way for variable-length denoising strategies, and we present **DAEDAL**, a novel training-free denoising strategy that enables **D**ynamic **A**daptive Length **E**xpansion for **D**iffusion **La**rge **L**anguage Models. DAEDAL operates in two phases: **1) Initial Length Adjustment.** Before the denoising process, DAEDAL starts from a short initial length and iteratively expands it to a coarse task-appropriate length. The expansion is guided by a sequence completion metric, which is calculated by the predicted confidence of EOS tokens within a fixed window. **2) Iterative Mask Insertion.** During the denoising process, DAEDAL dynamically expands the sequence to develop a better output. The expansion is realized by inserting mask tokens into insufficient regions, where the model struggles to plan and the corresponding prediction confidence is quite low.

With DAEDAL, DLLMs no longer require manually tuned, task-specific generation lengths. Instead, they can start from a short, unified initial length and dynamically expand as needed. Experiments

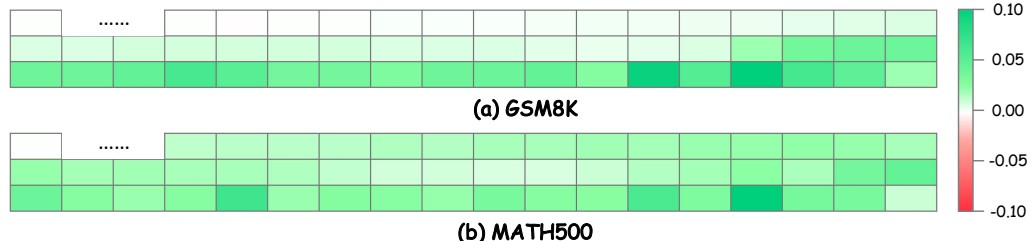

Figure 2: **Visualization of the DLLM's awareness of length sufficiency.** The heatmaps show the difference in average EOS token confidence at the sequence terminus, measured after the first prediction on a fully masked 128-token input. This difference is the result of subtracting the average confidence on **length-insufficient** problems (those answered correctly only with a much longer sequence) from that on **length-sufficient** problems (those answered correctly under 128 tokens). The experiment is conducted with LLaDA-Instruct-8B. The predominantly green color (difference $> 0$) indicates that EOS confidence is higher for length-sufficient problems, validating our core insight.

demonstrate that DAEDAL not only allocates appropriate computational resources for diverse tasks, as shown in Figure 1 (b), but also achieves performance that is comparable, and in some cases superior, to the peak performance of meticulously tuned fixed-length baselines. Our method thus achieves strong performance while significantly improving computational efficiency.

## 2 METHODS

### 2.1 OVERVIEW OF DIFFUSION LARGE LANGUAGE MODELS

**Training.** The training of a DLLM aims to define and learn a model distribution $p_\theta(x_0)$ that approximates the true data distribution. This is achieved through a forward and a reverse probabilistic process (Austin et al., 2021a; Ou et al., 2024; Shi et al., 2024). The forward process defines a fixed data noising mechanism indexed by a continuous time variable $t \in [0, 1]$. It progressively masks an original sequence $x_0$ until it is fully masked at $t = 1$. For any given $t$, a noised version $x_t$ is generated by independently replacing each token in $x_0$ with a [MASK] token with probability $t$, while keeping it unchanged with probability $1 - t$. The reverse process is where learning occurs. A Transformer model, parameterized by $\theta$, is trained to reverse the forward process by learning to predict the original sequence $x_0$ from its noised version $x_t$. This is achieved by optimizing the model to minimize a cross-entropy loss computed only on the masked token positions. This objective is principled as it corresponds to maximizing the Evidence Lower Bound (ELBO) on the data's log-likelihood, thereby pushing the model distribution $p_\theta$ to approximate the true data distribution.

**Inference.** During inference, LLaDA employs a multi-step iterative denoising process to generate text. As illustrated in Figure 3 (a), this process begins with a sequence of length $L$ composed entirely of [MASK] tokens, where $L$ is a predefined hyperparameter. In each denoising step $t$, the model takes the current sequence $\mathbf{x}_t$ as input and predicts the tokens for all masked positions, yielding an estimate $\hat{\mathbf{x}}_0$ of the complete, clean sequence. Subsequently, a "remasking" strategy is applied to determine which tokens are finalized and which are re-masked for the next denoising step, $t - 1$. LLaDA demonstrates the effectiveness of a "low-confidence remasking" strategy (Nie et al., 2025; Chang et al., 2022), where tokens predicted with the highest confidence are kept, while those with low confidence are re-masked for further refinement in subsequent steps. This iterative process continues for a fixed number of steps until the final text sequence is generated. This standard inference pipeline exposes a core problem: the generation length $L$ must be statically specified before inference begins, preventing it from adapting to the actual requirements of the task.

Further related works are provided in Appendix C.

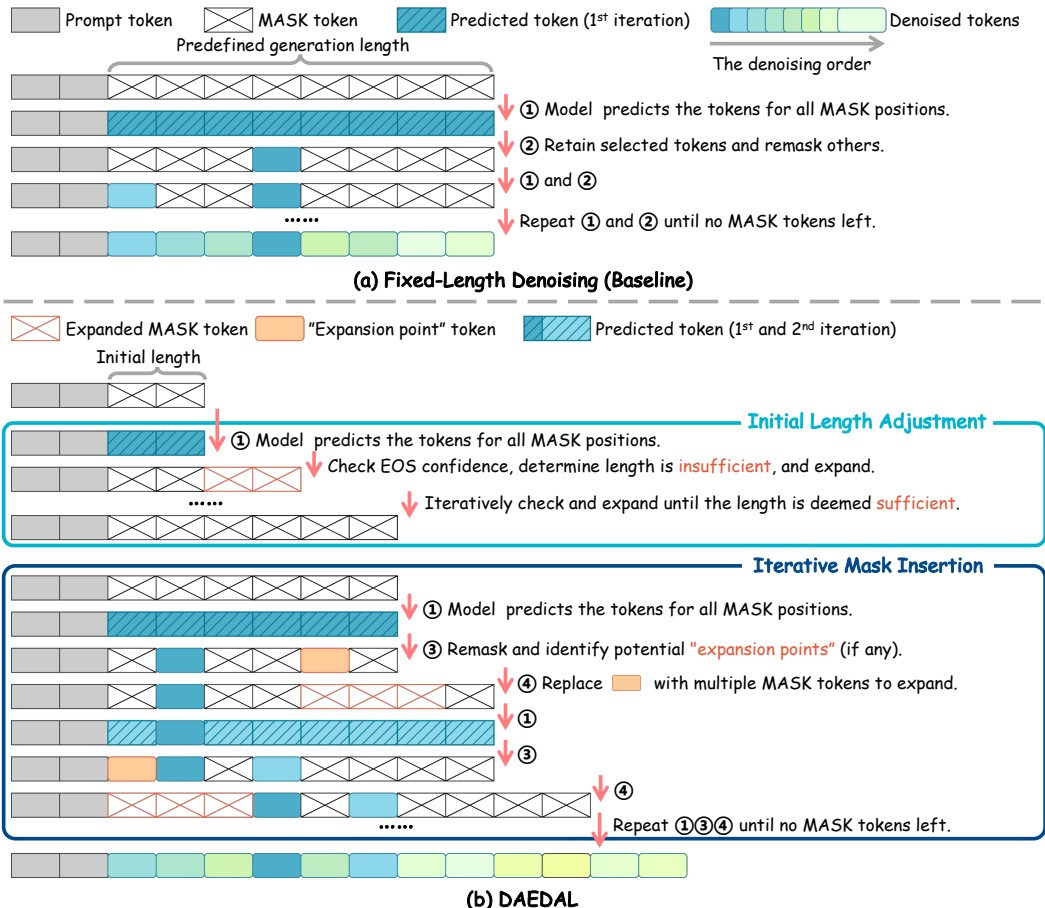

Figure 3: **Inference process of Fixed-Length Denoising (Baseline) and DAEDAL. (a)** The standard inference process for current DLLMs, which performs iterative denoising on a sequence of a predefined, static length. **(b)** Our proposed two-stage inference process, which first employs **Initial Length Adjustment** to determine an appropriate generation length before denoising, followed by **Iterative Mask Insertion** to expand the sequence on-demand during the denoising process.

## 2.2 DAEDAL

To address the static length limitation of standard DLLM inference, we introduce DAEDAL, a training-free, two-stage strategy. This approach allows the model to allocate an appropriate sequence length for each task and to insert additional space for reasoning where needed. A detailed algorithmic description of this process is provided in Algorithm 1 (Appendix D).

### 2.2.1 INITIAL LENGTH ADJUSTMENT

The first stage of DAEDAL is Initial Length Adjustment. The core insight of it is that the model's confidence in generating an End-of-Sequence (EOS) token at the end of the sequence can be interpreted as an internal signal of whether the current token length is sufficient. To validate this insight, we visualize the model's behavior in Figure 2. Specifically, we measure the average EOS confidence at the sequence terminus after the first prediction, when the predefined generation length is 128. We then compare this confidence between two empirically defined groups of problems: those answered correctly in under 128 tokens (length-sufficient) and those answered correctly only with a much longer sequence (length-insufficient). The visualization clearly shows that the model predicts EOS tokens with significantly higher confidence for the length-sufficient problems, indicating an awareness that the current length is adequate. Conversely, for problems where the length is insufficient, the model utilizes the available space more thoroughly, resulting in lower EOS confidence at the sequence's

terminus. If the model deems the current length inadequate to fully articulate its response, it will tend to utilize all available space, making it less likely to generate EOS tokens with high confidence.

Based on this insight, we introduce a preliminary length estimation loop that precedes the main denoising process. As depicted in Figure 3 (b), this loop begins with a short initial generation length. In each estimation iteration, the model performs a forward pass on the current sequence (prompt plus [MASK] tokens) to evaluate its predictions. We specifically focus on a window at the end of the sequence and calculate the model's average confidence in predicting the EOS token in these positions. If this confidence falls below a predefined threshold, we interpret this as a "length insufficient" signal. This indicates that the model, being forced to conclude prematurely, has not yet formed a complete response and is therefore unwilling to commit to an EOS token. In response, we expand the generation length by appending additional [MASK] tokens to the end of the sequence. This length adjustment loop repeats until the EOS confidence surpasses the threshold or a maximum length limit is reached. Through this mechanism, DAEDAL dynamically allocates a task-appropriate generation length for the model before commencing the fine-grained denoising process.

### 2.2.2 ITERATIVE MASK INSERTION

After the first stage allocates an appropriate length for the task, the second stage of DAEDAL, Iterative Mask Insertion, further enhances generation flexibility during the denoising process. We propose that predictions with exceptionally low confidence are not merely signals of uncertainty; on a deeper level, they indicate that the local context is too constrained to articulate a complex thought or logical step. In other words, this is a signal that the model requires more "discursive space" to refine its reasoning.

Therefore, during each denoising step, in addition to identifying and filling high-confidence tokens, our method also flags the masked position with the lowest prediction confidence, provided it falls below a very low threshold. As shown in Figure 3 (b), this position is not treated as merely a "difficult token" but is marked as an "expansion point". When a position is marked for expansion, instead of simply remasking it, we dynamically replace the single [MASK] token with a block of multiple [MASK] tokens. This operation effectively inserts additional space into the sequence.

This mechanism provides the model with "breathing room" precisely where complex reasoning or detailed description is needed, allowing it to better structure its language and logic in subsequent denoising iterations. Unlike the first stage, which performs a holistic length adjustment, Iterative Mask Insertion is a localized, on-demand refinement that occurs in real-time during generation. This enables DAEDAL to handle complex scenarios where the required length exceeds the initial length estimate, significantly enhancing the model's expressive ability.

## 3 EXPERIMENTS

### 3.1 IMPLEMENTATION DETAILS

We use LLaDA-Instruct-8B and LLaDA-1.5-8B as our baseline models. To demonstrate the generalization ability of DAEDAL, we also conducted experiments on Dream-Instruct-7B. For fairness, we did not use any acceleration or caching optimization methods proposed in subsequent works. **Crucially, DAEDAL employs the exact same hyperparameters across all experiments and models, with no model-specific tuning.** Detailed configurations are provided in Appendix E.

### 3.2 BENCHMARKS AND METRICS

To comprehensively evaluate the effectiveness of DAEDAL, we conducted experiments on four benchmarks spanning the domains of mathematical reasoning and code generation. For mathematical reasoning, we utilize GSM8K (Cobbe et al., 2021), which consists of grade-school math word problems to assess multi-step reasoning, and the more challenging MATH500 (Lightman et al., 2023) benchmark, composed of competition-level mathematics problems; performance on both is measured by accuracy. To evaluate code generation, we employ MBPP (Austin et al., 2021b) benchmark for entry-level Python tasks and the more complex, handwritten HumanEval (Chen et al., 2021) benchmark to test program synthesis capabilities. For these code generation tasks, we report the pass@1 metric to assess the correctness of the generated code in a single attempt.

Table 1: **Main Results of DAEDAL on LLaDA-Instruct-8B.** We compare the baseline performance at various generation lengths (64 to 2048) against DAEDAL. $Acc$ denotes accuracy, $E_{token}$ is the average effective tokens (the response length excluding trailing padding), $N_{token}$ is the average total tokens, and $E_{ratio}$ is the effective token ratio. The best configuration for the baseline is highlighted in orange. The **best** results are **bold and underlined**, and the second-best results are underlined.

| Benchmark | Metric | Fixed-Length Denoising (Baseline) | | | | | | DAEDAL |
|---|---|---|---|---|---|---|---|---|
| | | 64 | 128 | 256 | 512 | 1024 | 2048 | 64 |
| GSM8K | $Acc$ | 48.0 | 67.9 | 77.6 | 83.3 | 83.8 | 82.6 | **85.8** |
| | $E_{token}$ | 62 | 124 | 234 | 287 | 284 | 294 | 267 |
| | $N_{token}$ | 64 | 128 | 256 | 512 | 1024 | 2048 | 363 |
| | $E_{ratio}$ | 97.1% | 97.0% | 91.2% | 56.0% | 27.7% | 14.4% | 73.5% |
| MATH500 | $Acc$ | 24.0 | 29.0 | 35.6 | 38.8 | 39.4 | 39.6 | **44.2** |
| | $E_{token}$ | 62 | 123 | 245 | 424 | 583 | 718 | 541 |
| | $N_{token}$ | 64 | 128 | 256 | 512 | 1024 | 2048 | 704 |
| | $E_{ratio}$ | 96.4% | 96.4% | 95.8% | 82.8% | 56.9% | 35.1% | 76.8% |
| MBPP | $Acc$ | 20.8 | 28.0 | 37.4 | 38.2 | 37.4 | 38.8 | **40.8** |
| | $E_{token}$ | 61 | 122 | 232 | 331 | 335 | 336 | 324 |
| | $N_{token}$ | 64 | 128 | 256 | 512 | 1024 | 2048 | 618 |
| | $E_{ratio}$ | 95.1% | 95.7% | 90.6% | 64.7% | 32.7% | 16.4% | 52.5% |
| HUMANEVAL | $Acc$ | 18.9 | 26.2 | 36.0 | 47.0 | 47.6 | 47.0 | **48.2** |
| | $E_{token}$ | 60 | 125 | 245 | 471 | 641 | 669 | 523 |
| | $N_{token}$ | 64 | 128 | 256 | 512 | 1024 | 2048 | 813 |
| | $E_{ratio}$ | 93.2% | 97.6% | 95.6% | 92.0% | 62.6% | 32.7% | 64.3% |
| **Average** $Acc$ | | 27.93 | 37.78 | 46.65 | 51.83 | 52.05 | 52.00 | **54.75** |

## 3.3 MAIN RESULTS

We conducted a comprehensive evaluation on four benchmarks. The results for LLaDA-Instruct-8B, LLaDA-1.5-8B, and Dream-Instruct-7B, comparing the Fixed-Length Denoising baseline against our DAEDAL method, are presented in Table 1, Table 5 (Appendix F), and Table 6 (Appendix G), respectively. For the baseline models, performance is highly dependent on manually tuning the generation length for each specific task. We therefore report baseline performance across six fixed-length configurations, from 64 to 2048 tokens. In addition to accuracy ($Acc$), we introduce three key metrics: the total number of tokens generated ($N_{token}$), which for the baseline is its preset fixed length; the number of effective tokens used to answer the question ($E_{token}$), representing the "net" response length after removing trailing EOS padding; and the effective token ratio ($E_{ratio}$).

**DAEDAL achieves strong performance with a unified initial length**. The results clearly demonstrate the superior performance of DAEDAL. Despite starting with a short initial length, DAEDAL's two-stage length adjustment and expansion mechanism allows it to not only significantly outperform baselines with the same initial length, but also achieve performance that is comparable, and in some cases superior, to the peak performance of the meticulously tuned fixed-length baseline. This finding highlights the effectiveness of DAEDAL and exposes the inherent impracticality of the fixed-length paradigm, as the optimal length for the baseline varies across different benchmarks, underscoring the necessity of dynamic length adaptation. To visually illustrate this dynamic adaptability, Figure 4 contrasts the distribution of generation lengths ($N_{token}$) used by DAEDAL against the single fixed length of the best-performing baseline for each benchmark. Unlike the baseline, which is constrained to a pre-defined length for all problems, DAEDAL automatically adapts its generation length on a per-problem basis, resulting in a diverse length distribution that reflects varying task complexity.

**DAEDAL adaptively finds the optimal generation length.** Further analysis reveals that DAEDAL intelligently estimates and generates responses of an appropriate length. In most cases, the number

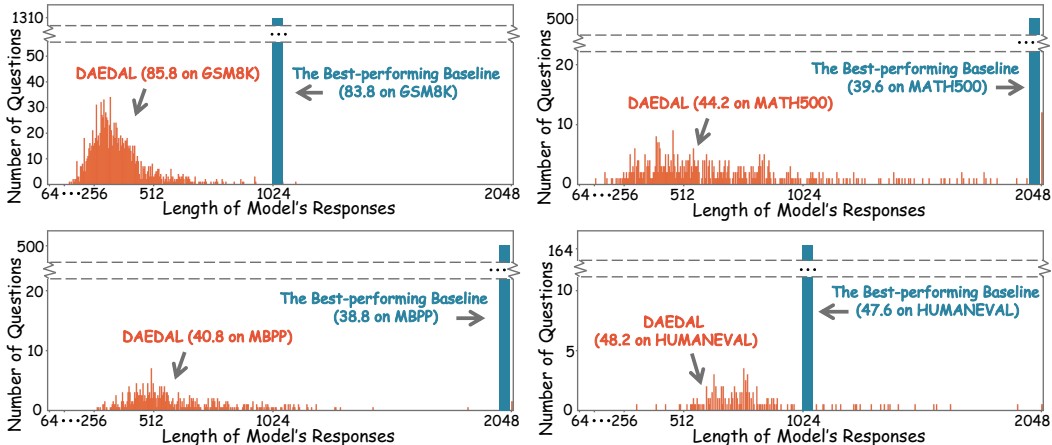

Figure 4: **Distribution of individual Response Lengths ($N_{token}$) on LLaDA-Instruct-8B.** The figure compares the distribution of total tokens used per problem by DAEDAL (orange histogram) and the baseline (blue histogram) across four benchmarks. DAEDAL's dynamic, per-problem adaptation results in a varied distribution of lengths. In contrast, the baseline is constrained to a single fixed length for all problems within a benchmark, represented by a single bar in its histogram.

of effective tokens ($E_{token}$) produced by DAEDAL is comparable to that of the baseline's best-performing configuration. This suggests that DAEDAL adaptively finds the model's inherent "sweet spot" for the token length required by a given task. The baseline's behavior corroborates this: when the fixed length is set beyond its optimal point, performance degrades even though the number of effective tokens may continue to increase. DAEDAL's adaptive nature effectively avoids this performance decay from over-expansion.

**DAEDAL significantly improves computational efficiency.** Furthermore, DAEDAL offers significant efficiency advantages. While achieving superior accuracy, the total number of tokens ($N_{token}$) generated by DAEDAL is generally lower than that of the baseline at its peak-performance setting. A similar effective token count with a lower total token count results in a much higher effective token ratio ($E_{ratio}$). This dramatically improves the utilization of the computational resource by reducing the overhead of bidirectional attention on unnecessarily long sequences and minimizing wasted resources on generating meaningless padding tokens.

## 3.4 ANALYSIS ON DAEDAL

**DAEDAL's two stages are individually effective and synergistic when combined.** As shown in Table 2, we conducted an ablation study to analyze the individual contributions of DAEDAL's two core components: Initial Length Adjustment (Stage 1 only) and Iterative Mask Insertion (Stage 2 only). The results indicate that applying either stage individually yields significant performance gains over the baseline. The full DAEDAL method, which combines both stages, ultimately achieves the best performance, surpassing the results of using either stage alone. This demonstrates that the two stages are complementary and indispensable for achieving optimal results.

**DAEDAL's stages highlights the importance of the initial length for global planning.** When using the Iterative Mask Insertion (Stage 2) stage in isolation, we observe that its performance is sensitive to the initial length. When starting from a very short initial length (e.g., 64), its performance, while substantially better than the baseline at that same length, still falls short of the baseline's peak performance achieved at an optimal, longer length. Yet, when initiated with a more reasonable length (e.g., 256), this stage alone can surpass the baseline's overall best result. This behavior is expected and highlights the nature of the second stage as a local, on-demand expansion mechanism. If the initial length is severely constrained, the DLLM's ability to form a sound global plan for the solution is compromised from the outset. While subsequent local expansions can provide remedies, the overall performance ceiling is still limited by the flawed initial plan. This not only demonstrates the effectiveness of Iterative Mask Insertion but also underscores the necessity of Initial Length Adjustment (Stage 1) for establishing a solid foundation for global planning.

Table 2: **Ablation Results of DAEDAL's Two Stages.** Experiments are conducted on GSM8K with LLaDA-Instruct-8B. We compare the performance of the full DAEDAL method, as well as its individual stages (Stage 1 and Stage 2), against the baseline. The baseline is evaluated at multiple fixed lengths, with its best configuration highlighted in orange. Stage 1 and DAEDAL evaluated with an initial length of 64, while Stage 2 is evaluated with varying initial lengths (64, 128, 256).

| Metric | Fixed-Length Denoising (Baseline) | | | | | | w/ Stage1 | w/ Stage2 | | | DAEDAL |
|---|---|---|---|---|---|---|---|---|---|---|---|
| | 64 | 128 | 256 | 512 | 1024 | 2048 | 64 | 64 | 128 | 256 | 64 |
| $Acc$ | 48.0 | 67.9 | 77.6 | 83.3 | 83.8 | 82.6 | 84.1 | 72.3 | 81.1 | 84.7 | 85.8 |
| $E_{token}$ | 62 | 124 | 234 | 287 | 284 | 294 | 253 | 127 | 167 | 256 | 267 |
| $N_{token}$ | 64 | 128 | 256 | 512 | 1024 | 2048 | 311 | 152 | 209 | 340 | 363 |
| $E_{ratio}$ | 97.1% | 97.0% | 91.2% | 56.0% | 27.7% | 14.4% | 81.3% | 83.1% | 79.7% | 75.3% | 73.5% |

Table 3: **Ablation Results on DAEDAL's Initial Length.** Experiments are conducted on GSM8K and HUMANEVAL using LLaDA-Instruct-8B. DAEDAL is evaluated with initial lengths ranging from 32 to 512. We highlight our default setting ($L_{init} = 64$) in blue.

| Metric | GSM8K | | | | | HUMANEVAL | | | | |
|---|---|---|---|---|---|---|---|---|---|---|
| | 32 | 64 | 128 | 256 | 512 | 32 | 64 | 128 | 256 | 512 |
| $Acc$ | 85.8 | 85.8 | 85.8 | 85.8 | 85.1 | 48.2 | 48.2 | 48.2 | 48.2 | 48.2 |
| $E_{token}$ | 267 | 267 | 267 | 268 | 277 | 523 | 523 | 523 | 523 | 523 |
| $N_{token}$ | 363 | 363 | 363 | 369 | 532 | 813 | 813 | 813 | 813 | 816 |
| $E_{ratio}$ | 73.5% | 73.5% | 73.5% | 72.7% | 52.0% | 64.4% | 64.4% | 64.4% | 64.4% | 64.1% |

Table 4: **Ablation Results on DAEDAL's Expansion Factor and EOS Confidence Window Size.** Both ablation studies are conducted on GSM8K using LLaDA-Instruct-8B. The left panel shows results for varying $E_{factor}$ ranging from 8 to 32, and the right panel for varying $W_{eos}$ ranging from 8 to 32. We highlight our default setting ($E_{factor} = 8$ and $W_{eos}$=32) in blue.

| Metric | Expansion Factor ($E_{factor}$) | | | | | Metric | EOS Confidence Window Size ($W_{eos}$) | | | |
|---|---|---|---|---|---|---|---|---|---|---|
| | 8 | 16 | 24 | 32 | | | 8 | 16 | 24 | 32 |
| $Acc$ | 85.8 | 85.8 | 86.4 | 86.3 | | $Acc$ | 82.9 | 83.5 | 84.4 | 85.8 |
| $E_{token}$ | 267 | 272 | 272 | 274 | | $E_{token}$ | 232 | 252 | 260 | 267 |
| $N_{token}$ | 363 | 386 | 392 | 405 | | $N_{token}$ | 289 | 327 | 347 | 363 |
| $E_{ratio}$ | 73.5% | 70.5% | 69.3% | 67.6% | | $E_{ratio}$ | 80.2% | 77.1% | 75.0% | 73.5% |

**DAEDAL exhibits strong robustness to the initial length.** A core advantage of DAEDAL is its ability to achieve strong performance from a short, unified initial length. We conduct an ablation study to verify its robustness to this hyperparameter, $L_{init}$. As shown in Table 3, DAEDAL delivers remarkably stable performance across a wide range of initial lengths, from 32 to 512 tokens. On HumanEval, the accuracy remains identical across all settings, while the variation on GSM8K is negligible. This result demonstrates that DAEDAL is largely insensitive to the initial length setting. It confirms that users do not need to meticulously tune this hyperparameter; a unified and short initial length (e.g., 64) is sufficient to achieve optimal performance, fulfilling its original design objective.

**DAEDAL's performance is insensitive to the expansion factor granularity.** We conduct an ablation study on the expansion factor, which controls the number of [MASK] tokens added during a single expansion event. As shown in the left panel of Table 4, DAEDAL's performance remains remarkably stable across a range of expansion factors (8 to 32). This result suggests that the specific granularity of each expansion step is not critical. A smaller factor leads to more frequent, fine-grained expansions, while a larger factor results in fewer, coarser expansions. Regardless of the step size, the model robustly converges to a similar, task-appropriate total length. This finding further corroborates

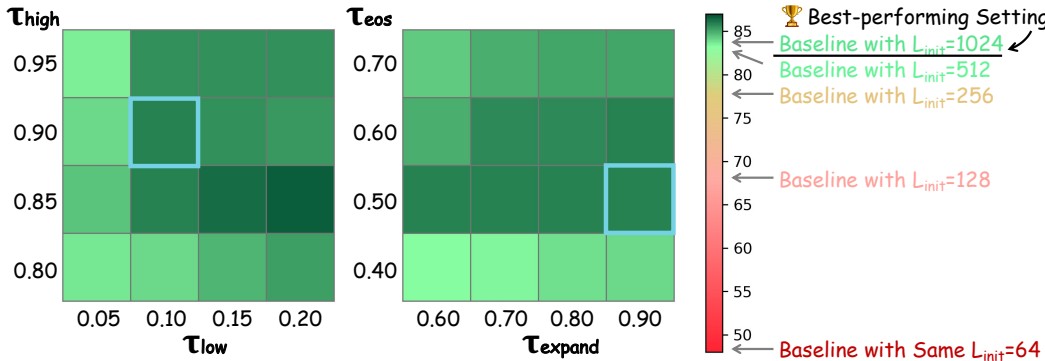

Figure 5: **Ablation Results on DAEDAL's Thresholds.** The two 4x4 heatmaps present a grid search over two interdependent threshold pairs: $(\tau_{high}, \tau_{low})$ and $(\tau_{eos}, \tau_{expand})$. All 32 configurations were evaluated on GSM8K using LLaDA-Instruct-8B. Higher accuracy is indicated by a darker green. The color bar also provides reference color for performance of baseline. Our default settings are in **blue** boxes. The results demonstrate remarkable stability, with all configurations comparable to the best-performing baseline, and some even outperforming it.

our core hypothesis that the model possesses a strong internal estimate of the length required for a given problem, and DAEDAL's mechanism effectively enables the model to reach that target.

**DAEDAL is robust to the EOS confidence window size, achieving optimal performance with a large window.** We analyze the sensitivity to the EOS confidence window size, used to determine length sufficiency. As shown in the right panel of Table 4, performance is stable for larger window sizes but degrades for very small values. Notably, even this sub-optimal configuration still yields significant gains over the baseline at a comparable short initial length (77.3 vs. 48.0 Acc on GSM8K when $L_{init}$=64). We attribute the performance drop at small window sizes to a higher probability of misjudging the model's intent. A larger window provides a more robust signal by averaging confidence over a wider context. In contrast, a small window is susceptible to localized fluctuations and may prematurely terminate length expansion based on a few high-confidence EOS tokens at the sequence's immediate end, leading to length under-allocation and a drop in final performance.

**DAEDAL demonstrates broad robustness across its threshold settings.** We conduct a comprehensive ablation study on all the four key threshold hyperparameters in DAEDAL: $\tau_{eos}, \tau_{expand}, \tau_{high}, \tau_{low}$. We analyze these in interdependent pairs via a grid search: $(\tau_{high}, \tau_{low})$, which govern token-level filling and expansion decisions. Specifically, the high-confidence threshold $\tau_{high}$ cis analogous to the confident decoding strategy proposed in Dimple, which aims to accelerate inference by simultaneously filling all tokens that exceed a certain confidence level. The second pair, $(\tau_{eos}, \tau_{expand})$, controls sequence-level length adjustments. The results on GSM8K, presented in Figure 5, demonstrate DAEDAL's exceptional robustness. Across the 32 tested configurations, all configurations are comparable to the best-performing baseline (83.8 Acc), with some even outperforming it, and the overall performance variation across all settings is minimal. This indicates that DAEDAL is largely insensitive to the precise choice of these thresholds, confirming that it can deliver strong and stable performance without requiring extensive hyperparameter tuning.

# 4 CONCLUSION

In this work, we addressed a fundamental architectural constraint of Diffusion Large Language Models (DLLMs): the reliance on a statically predefined generation length. This limitation hinders their practical application by creating a difficult dilemma: insufficient lengths cripple performance on complex tasks, while excessive lengths not only incur significant computational overhead but can sometimes lead to performance degradation. We introduced DAEDAL, a novel training-free, two-stage denoising strategy that resolves this issue by leveraging the model's own internal signals. DAEDAL first performs an Initial Length Adjustment to set a coarse, task-appropriate budget, and then uses Iterative Mask Insertion to dynamically expand the sequence at regions requiring more detailed reasoning during the denoising process. Our extensive experiments and analyses demonstrate that

DAEDAL successfully endows DLLMs with the ability for dynamic, per-problem length adaptation. This allows a model starting from a short, unified initial length to achieve performance that is comparable, and at times superior, to that of meticulously tuned fixed-length baselines. By removing the need for manual length tuning and enabling the model to find its own optimal response length, DAEDAL not only enhances performance but also improves computational efficiency. Ultimately, this work bridges a critical capability gap between diffusion and autoregressive models, paving the way for more flexible, efficient, and capable non-autoregressive language generation.

ACKNOWLEDGMENTS

This project is funded in part by Shanghai Artificial Intelligence Laboratory, Shanghai Innovation Institute, the Centre for Perceptual and Interactive Intelligence (CPII) Ltd under the Innovation and Technology Commission (ITC)'s InnoHK. Dahua Lin is a PI of CPII under the InnoHK.

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

## A    Reproducibility Statement

To ensure the reproducibility of our work, we provide a detailed algorithmic description of DAEDAL in Appendix D. All hyperparameters and hardware specifications used in our experiments are explicitly listed in Appendix E. The source code for reproducing our main results has been included in the Supplementary Materials.

## B    The Use of Large Language Models

We utilized Large Language Models (LLMs), such as Google's Gemini, as a writing assistance tool. The use of the LLM was strictly limited to improving the clarity, conciseness, and grammatical correctness of the text. Specific applications included proofreading for typographical errors, rephrasing complex sentences to enhance readability, and ensuring consistent terminology throughout the paper.

## C    Related Works

**Diffusion Large Language Models.**    In recent years, Diffusion Language Models (DLLMs) have emerged as a prominent area of research. Among them, LLaDA (Nie et al., 2025) stands out as the first large-scale diffusion model trained from scratch to reach the billion-parameter scale. Trained on 2.3 trillion tokens, LLaDA-8B has demonstrated performance competitive with state-of-the-art autoregressive models like LLaMA-3-8B (Grattafiori et al., 2024). Indeed, autoregressive architectures have long served as the dominant foundation across diverse modalities, ranging from standard text generation to complex streaming video understanding and real-time interaction (Zhang et al., 2023; Qian et al., 2024a; Ding et al., 2024; Qian et al., 2024b; Zhang et al., 2024; Qian et al., 2025). LLaDA proves the remarkable scalability and potential of the native diffusion architecture on multiple tasks (Hendrycks et al., 2020; Suzgun et al., 2022; Cobbe et al., 2021). Subsequently, LLaDA-1.5 (Zhu et al., 2025) advanced this paradigm by successfully applying reinforcement learning for preference alignment, achieving significant further improvements on benchmarks for mathematics, code, and alignment. In contrast to LLaDA, another line of research has explored adapting existing AR LLMs into DLLMs. For instance, models like DiffuLLaMA (Gong et al., 2024) and Dream (Ye et al., 2025) were developed by fine-tuning pre-trained AR LLMs such as GPT2 (Radford et al., 2019), LLaMA2 (Touvron et al., 2023) and Qwen (Yang et al., 2024). While these adapted models have also achieved strong results, our work focuses more on native, from-scratch DLLMs like LLaDA, seeking to explore their generation mechanisms and address the specific challenge of fixed-length inference.

**Inference Strategies for DLLMs.** Existing research on inference strategies for DLLMs has predominantly focused on enhancing generation speed through computational optimizations. For example, Fast-dLLM (Wu et al., 2025) introduces a novel block-wise approximate Key-Value (KV) Cache tailored for bidirectional attention models, combined with a confidence-aware parallel decoding strategy, to achieve up to an improvement in throughput. Similarly, dLLM-Cache (Liu et al., 2025) observes the static nature of prompts and the dynamic sparsity of responses during DLLM inference, proposing an adaptive caching framework that combines long-interval prompt caching with partial response updates to achieve lossless speedup. Dimple (Yu et al., 2025) proposes a "Confident Decoding" strategy that dynamically adjusts the number of tokens generated at each step based on model confidence, thereby significantly reducing the total number of iterations. In summary, while all these methods (Ma et al., 2025; Israel et al., 2025; Ben-Hamu et al., 2025) have made significant strides in improving the inference speed of DLLMs via computational caching and parallel decoding. They do not address the more fundamental issue that the generation length itself needs to adapt dynamically to different task requirements. To our knowledge, the problem of dynamically adjusting and expanding the total generation length of DLLMs at inference time remains unexplored. Our work, therefore, aims to fill this critical gap by proposing a novel dynamic adaptive expansion strategy.

## D    PSEUDO-CODE OF DAEDAL

---

**Algorithm 1** The DAEDAL Inference Process

---

1: **Input:** Prompt $\mathbf{c}$, model $f_\theta$, initial/max length $L_{init}/L_{max}$, thresholds $\tau_{eos}, \tau_{high}, \tau_{low}, \tau_{expand}$, expansion factor $E_{factor}$, EOS confidence window size $W_{eos}$

2: **Output:** Generated sequence $\mathbf{y}$

                 ▷ **Stage 1: Initial Length Adjustment**

3: $\mathbf{x} \leftarrow [\mathbf{c}, \underbrace{[\texttt{MASK}], \ldots, [\texttt{MASK}]}_{L_{init}}]$          ▷ Initialize sequence

4: **while** length$(\mathbf{x}) < L_{max}$ **do**

5:    $L_{\text{logits}} \leftarrow f_\theta(\mathbf{x})$

6:    $\text{conf}_{\text{eos}} \leftarrow \text{ComputeEOSConfidence}(L_{\text{logits}}, \mathbf{x}, W_{eos})$

7:    **if** $\text{conf}_{\text{eos}} < \tau_{eos}$ **then**

8:      $\mathbf{x} \leftarrow [\mathbf{x}, \underbrace{[\texttt{MASK}], \ldots, [\texttt{MASK}]}_{E_{factor}}]$    ▷ Expand length if EOS confidence is low

9:    **else**

10:      **break**             ▷ length is sufficient, exit loop

11:    **end if**

12: **end while**

             ▷ **Stage 2: Iterative Denoising and Mask Insertion**

13: **while** ContainsMask$(\mathbf{x})$ **do**

14:    $L_{\text{logits}} \leftarrow f_\theta(\mathbf{x})$

15:    $P_{\text{conf}}, \hat{\mathbf{x}} \leftarrow \text{GetConfidenceAndPredictions}(L_{\text{logits}})$

16:    $\mathcal{M}_{\text{masked}} \leftarrow \{i \mid \mathbf{x}_i = [\texttt{MASK}]\}$

17:    $\mathcal{I}_{\text{fill}} \leftarrow \{i \mid i \in \mathcal{M}_{\text{masked}} \wedge P_{\text{conf},i} > \tau_{high}\}$    ▷ Identify high-confidence set for filling

18:    $\mathcal{I}_{\text{candidates}} \leftarrow \{i \mid i \in \mathcal{M}_{\text{masked}} \wedge P_{\text{conf},i} < \tau_{low}\}$   ▷ Find low-confidence candidate positions

19:    **for** $j \in \mathcal{I}_{\text{fill}}$ **do**

20:      $\mathbf{x}_j \leftarrow \hat{\mathbf{x}}_j$           ▷ Fill in high-confidence tokens

21:    **end for**

22:    $\text{conf}_{\text{eos}} \leftarrow \text{ComputeEOSConfidence}(L_{\text{logits}}, \mathbf{x}, W_{eos})$

23:    **if** $\text{conf}_{\text{eos}} < \tau_{expand} \wedge \text{length}(\mathbf{x}) < L_{max} \wedge |\mathcal{I}_{\text{candidates}}| > 0$ **then**

24:      $i_{\text{expand}} \leftarrow \arg\min_{i \in \mathcal{I}_{\text{candidates}}} P_{\text{conf},i}$    ▷ Select the position with the lowest confidence

25:      Replace $\mathbf{x}_{i_{\text{expand}}}$ with $[\underbrace{[\texttt{MASK}], \ldots, [\texttt{MASK}]}_{E_{factor}}]$ ▷ Expand sequence at the selected position

26:    **end if**

27: **end while**

28: **return** $\mathbf{x}$

---

## E    EXPERIMENTS CONFIGURATION

All experiments were conducted on a server equipped with 8 NVIDIA A800 80G GPUs with the batch size set to 8. We employ a single, universal set of hyperparameters across all experiments and models, requiring no model-specific tuning. Specifically, the initial and maximum lengths are set to $L_{init} = 64$ and $L_{max} = 2048$. The thresholds are configured as $\tau_{eos} = 0.5$, $\tau_{high} = 0.9$, $\tau_{low} = 0.1$, and $\tau_{expand} = 0.9$. The expansion factor is $E_{factor} = 8$, and the EOS confidence window size is $W_{eos} = 32$. We also demonstrate DAEDAL's robustness to these choices in Section 3.4, which provides a detailed ablation study on these hyperparameters.

# F    MAIN RESULTS OF DAEDAL ON LLaDA-1.5-8B

Table 5: **Main Results of DAEDAL on LLaDA-1.5-8B.** We compare the baseline performance at various generation lengths (64 to 2048) against DAEDAL. $Acc$ denotes accuracy, $E_{token}$ is the average effective tokens (the response length excluding trailing padding), $N_{token}$ is the average total tokens, and $E_{ratio}$ is the effective token ratio. The best configuration for the baseline is highlighted in orange. The **best** results are **bold and underlined**, and the second-best results are underlined.

| Benchmark | Metric | Fixed-Length Denoising (Baseline) | | | | | | DAEDAL |
|---|---|---|---|---|---|---|---|---|
| | | 64 | 128 | 256 | 512 | 1024 | 2048 | 64 |
| GSM8K | $Acc$ | 49.4 | 71.0 | 80.4 | 83.7 | 83.8 | 84.5 | **85.5** |
| | $E_{token}$ | 62 | 125 | 237 | 293 | 287 | 292.5 | 275 |
| | $N_{token}$ | 64 | 128 | 256 | 512 | 1024 | 2048 | 377 |
| | $E_{ratio}$ | 97.4% | 97.5% | 92.6% | 57.2% | 28.1% | 4.3% | 73.0% |
| MATH500 | $Acc$ | 23.2 | 29.6 | 35.4 | 40.2 | **43.6** | 40.2 | 42.4 |
| | $E_{token}$ | 62 | 125 | 246 | 429 | 584 | 717 | 588 |
| | $N_{token}$ | 64 | 128 | 256 | 512 | 1024 | 2048 | 743 |
| | $E_{ratio}$ | 96.9% | 97.5% | 96.2% | 83.7% | 57.1% | 35.0% | 75.2% |
| MBPP | $Acc$ | 20.6 | 30.2 | 39.2 | 38.6 | 39.8 | 39.6 | **40.2** |
| | $E_{token}$ | 61 | 124 | 237 | 352 | 344 | 356 | 342 |
| | $N_{token}$ | 64 | 128 | 256 | 512 | 1024 | 2048 | 645 |
| | $E_{ratio}$ | 95.1% | 96.6% | 92.7% | 68.7% | 33.6% | 17.4% | 53.0% |
| HUMANEVAL | $Acc$ | 18.3 | 22.0 | 37.8 | 45.1 | 49.4 | **50.0** | 48.8 |
| | $E_{token}$ | 60 | 125 | 251 | 475 | 677 | 754 | 561 |
| | $N_{token}$ | 64 | 128 | 256 | 512 | 1024 | 2048 | 848 |
| | $E_{ratio}$ | 94.0% | 97.7% | 98.2% | 92.9% | 66.1% | 36.8% | 66.2% |
| Average $Acc$ | | 27.88 | 38.20 | 48.20 | 51.90 | 54.15 | 53.58 | **54.23** |

# G  MAIN RESULTS OF DAEDAL ON DREAM-INSTRUCT-7B

Table 6: **Main Results of DAEDAL on Dream-Instruct-7B.** We compare the baseline performance at various generation lengths (64 to 2048) against DAEDAL. $Acc$ denotes accuracy, $E_{token}$ is the average effective tokens (the response length excluding trailing padding), $N_{token}$ is the average total tokens, and $E_{ratio}$ is the effective token ratio. The best configuration for the baseline is highlighted in orange. The **best** results are **bold and underlined**, and the second-best results are underlined.

| Benchmark | Metric | Fixed-Length Denoising (Baseline) | | | | | | DAEDAL |
|---|---|---|---|---|---|---|---|---|
| | | 64 | 128 | 256 | 512 | 1024 | 2048 | 64 |
| GSM8K | $Acc$ | 20.8 | 51.9 | 81.3 | 83.7 | 83.8 | 83.7 | **84.2** |
| | $E_{token}$ | 63 | 123 | 202 | 217 | 217 | 232 | 209 |
| | $N_{token}$ | 64 | 128 | 256 | 512 | 1024 | 2048 | 326 |
| | $E_{ratio}$ | 98.0% | 96.3% | 78.9% | 42.3% | 21.2% | 11.3% | 63.9% |
| MATH500 | $Acc$ | 15.2 | 30.6 | 40.8 | 44.8 | 45.2 | 44.0 | **49.2** |
| | $E_{token}$ | 62 | 122 | 223 | 334 | 422 | 632 | 371 |
| | $N_{token}$ | 64 | 128 | 256 | 512 | 1024 | 2048 | 540 |
| | $E_{ratio}$ | 97.5% | 95.0% | 87.2% | 65.3% | 41.2% | 30.8% | 68.7% |
| MBPP | $Acc$ | 25.6 | 39.6 | **50.4** | 47.4 | 49.0 | 50.0 | 49.4 |
| | $E_{token}$ | 60.3 | 116.2 | 182.4 | 228.5 | 252.2 | 294.4 | 190.7 |
| | $N_{token}$ | 64 | 128 | 256 | 512 | 1024 | 2048 | 493 |
| | $E_{ratio}$ | 94.2% | 90.8% | 71.3% | 44.6% | 24.6% | 14.4% | 38.7% |
| HUMANEVAL | $Acc$ | 7.3 | 30.5 | 51.8 | **57.3** | 51.8 | 55.5 | 53.7 |
| | $E_{token}$ | 48 | 123 | 196 | 269 | 284 | 338 | 272 |
| | $N_{token}$ | 64 | 128 | 256 | 512 | 1024 | 2048 | 572 |
| | $E_{ratio}$ | 75.5% | 96.2% | 76.8% | 52.6% | 27.7% | 16.5% | 47.5% |
| **Average $Acc$** | | 17.22 | 38.14 | 56.08 | 58.31 | 57.45 | 58.30 | **59.12** |

# H    VISUALIZATIONS OF THE DLLM'S AWARENESS OF LENGTH SUFFICIENCY

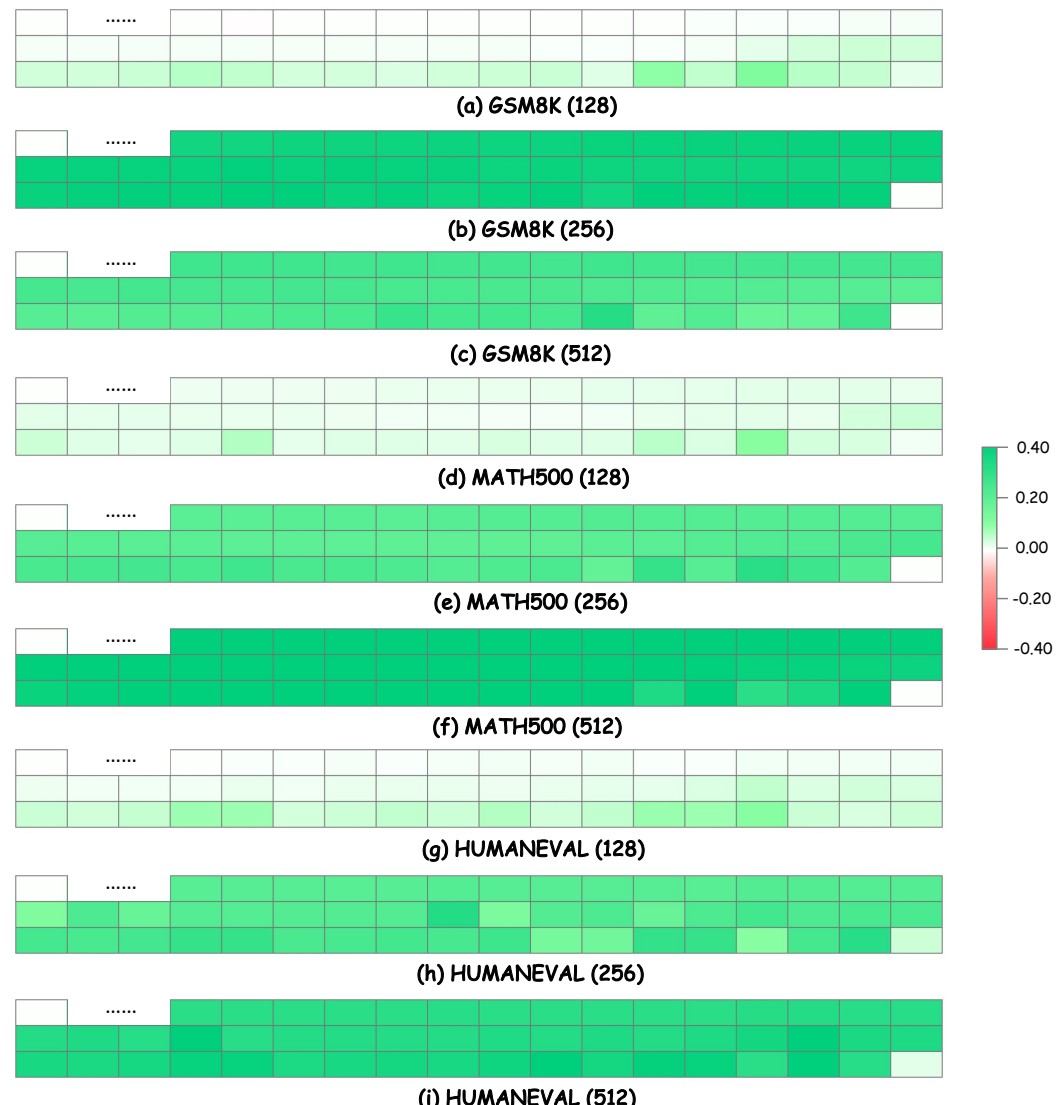

Figure 6: **Visualization of the DLLM's awareness of length sufficiency.** The heatmaps show the difference in average EOS token confidence at the sequence terminus, measured after the first prediction on a fully masked token input (length range from 128 to 512). This difference is the result of subtracting the average confidence on **length-insufficient** problems (those answered correctly only with a much longer sequence) from that on **length-sufficient** problems. The predominantly green color (difference > 0) indicates that EOS confidence is higher for length-sufficient problems, validating our core insight.

