# OpenReview forum: "Beyond Fixed: Training-Free Variable-Length Denoising for Diffusion Large Language Models"
_ICLR.cc/2026/Conference — ICLR 2026 Poster_

### Official Review · Reviewer_XwxZ · 2025-10-31

**Soundness:** 2
**Presentation:** 3
**Contribution:** 3
**Rating:** 6
**Confidence:** 3

**Summary:**

This paper tackles a central limitation of Diffusion LLMs—the need to pre-set a fixed output length—and proposes DAEDAL, a training-free, two-stage inference method that first estimates a task-appropriate length by checking EOS confidence at the sequence end and then expands low-confidence regions on the fly via mask insertion during denoising. On LLaDA-Instruct-8B and related DLLMs, DAEDAL matches or exceeds carefully tuned fixed-length baselines while using tokens more efficiently (e.g., GSM8K 85.8 vs. 83.8 accuracy with a much higher effective-token ratio), and extensive ablations show robustness to initial length, thresholds, and expansion factors.

**Strengths:**

- The proposed method is simple and intuitively reasonable. Requiring no retraining is a plus
- The proposed method demonstrates solid empirical improvements over the best-tuned fixed-length baselines on math reasoning and code generation benchmarks (e.g., MATH500), while generating effective tokens more efficiently.
- This paper conducts a thorough analysis of key hyperparameters of the method, showing its robustness to different configurations.

**Weaknesses:**

- While the paper shows in experiments that a combination of the two expansion stages gives the best performance, in principle, there still lacks a clear reason why both stages are necessary. It's natural to consider merging the first length adjustment stage into the second dynamic expansion stage. Interestingly, according to Table 2, stage 1 already contributes to most of the performance improvement. I think more investigation should be put into this.
- The ablation study shows the method's robustness to threshold hyperparameters, but it's only for a single model-dataset pair, i.e., LLaDA-Instruct-8B on GSM8K. I wonder whether the threshold findings can be transferred to other model-dataset combinations. Or do you need to tune thresholds again for a different dataset?
- It would be better to have some more direct measurements on the actual inference speed.
- The benchmarks focus on math and code. How would the proposed method perform on general language tasks?

**Questions:**

Please see Weaknesses

---

> ### Author Response · Authors · 2025-11-21
>
> We sincerely thank Reviewer XwxZ for the insightful summary and strong positive feedback. We are particularly encouraged that Reviewer XwxZ found the method **"simple and intuitively reasonable"** and recognized the **"solid empirical improvements"** and the **"thorough analysis of key hyperparameters"**.
>
> ---
>
> **Weakness 1: While the paper shows in experiments that a combination of the two expansion stages gives the best performance, in principle, there still lacks a clear reason why both stages are necessary. It's natural to consider merging the first length adjustment stage into the second dynamic expansion stage. Interestingly, according to Table 2, stage 1 already contributes to most of the performance improvement. I think more investigation should be put into this.**
>
> **Response 1:** This is a fantastic question that highlights the core architectural difference between global and local planning in DLLMs. The distinction lies in the crucial role of the initial sequence length for **Global Planning**. An overly constrained initial length (e.g., $L_{init}=64$), without Stage 1, fundamentally compromises the model's ability to form a sound solution structure (the "global plan") from the very beginning. Stage 1 (Initial Length Adjustment) is crucial because it ensures the initial input canvas is large enough for the model to establish a coherent, high level plan for the entire response *before* the detailed denoising process begins. Stage 2 (Iterative Mask Insertion) is purely a **local refinement** and **self correction** mechanism. It addresses instances where the initial global plan finds insufficient space for complex reasoning steps or detailed descriptions. As shown in Table 2, applying Stage 2 in isolation with a very short length ($L_{init}=64$, Acc 72.3%) significantly underperforms the full DAEDAL (Acc 85.8%), proving that local expansion cannot fully recover performance if the **global planning** framework was flawed from the outset. Thus, Stage 1 provides the necessary global foundation, and Stage 2 provides the necessary local flexibility, making them complementary and indispensable.
>
> ---
>
> **Weakness 2: The ablation study shows the method's robustness to threshold hyperparameters, but it's only for a single model-dataset pair, i.e., LLaDA-Instruct-8B on GSM8K. I wonder whether the threshold findings can be transferred to other model-dataset combinations. Or do you need to tune thresholds again for a different dataset?**
>
> **Response 2:** This is an important question regarding the generalization of our method. We want to explicitly state that **the same single set of hyperparameters is used across all models and benchmarks in our paper, with no task specific tuning**. The configurations are applied universally to LLaDA-Instruct-8B, LLaDA-1.5-8B (Table 5), and Dream-Instruct-7B (Table 6). The broad robustness demonstrated across four diverse benchmarks and three different DLLM models with **zero change in configuration** validates that the threshold findings are universally transferable. The model's awareness of length is an **intrinsic capability** guided by its prediction confidence. Therefore, DAEDAL's thresholds merely need to be set to a **reasonably chosen value** to utilize this robust signal, confirming that it can deliver strong and stable performance without requiring extensive hyperparameter tuning.
>
> ---

---

> > ### Author Response · Authors · 2025-11-21
> >
> > **Weakness 3: It would be better to have some more direct measurements on the actual inference speed.**
> >
> > **Response 3:** Thank you for this crucial suggestion. We have conducted wall-clock time measurements on **8 NVIDIA H800 GPUs**. We report the total computation time (batch size = 1) required for each fixed-length baseline configuration and for DAEDAL.
> >
> > * On GSM8K, to achieve a performance (83.8%) comparable to DAEDAL (85.8%), the baseline requires **11,545 seconds** of total computation. In contrast, DAEDAL achieves superior accuracy using only **1,049 seconds**, representing an **~11$\times$ speedup** in total computation time.
> >
> > * On MATH500, the baseline requires **14,196 seconds** for peak accuracy (39.6%), while DAEDAL achieves higher accuracy (44.2%) using only **1,160 seconds**, representing a **~12$\times$ speedup** in total computation time.
> >
> > | Benchmark | Method | Length | Acc | Avg Tokens | Total Time (s) |
> > | :--- | :--- | :--- | :--- | :--- | :--- |
> > | **GSM8K** | Baseline | 64 | 48.0 | 64 | 407 |
> > | | Baseline | 128 | 67.9 | 128 | 837 |
> > | | Baseline | 256 | 77.6 | 256 | 1825 |
> > | | Baseline | 512 | 83.3 | 512 | 4405 |
> > | | **Baseline** | **1024 (Peak Acc)** | **83.8** | 1024 | 11545 |
> > | | Baseline | 2048 | 82.6 | 2048 | 37016 |
> > | | **DAEDAL** | **Adaptive** | **85.8** | 363 | **1049** |
> > | **MATH500** | Baseline | 64 | 24.0 | 64 | 177 |
> > | | Baseline | 128 | 29.0 | 128 | 340 |
> > | | Baseline | 256 | 35.6 | 256 | 717 |
> > | | Baseline | 512 | 38.8 | 512 | 1690 |
> > | | Baseline | 1024 | 39.4 | 1024 | 4444 |
> > | | **Baseline** | **2048 (Peak Acc)** | **39.6** | 2048 | 14196 |
> > | | **DAEDAL** | **Adaptive** | **44.2** | 704 | **1160** |
> >
> > ---
> >
> > **Weakness 4: The benchmarks focus on math and code. How would the proposed method perform on general language tasks?**
> >
> > **Response 4:** Our current evaluation focuses on tasks where the required reasoning length varies significantly, which is key for validating a variable length mechanism. However, the core mechanism of DAEDAL relies on the model's **intrinsic confidence signal**, which is universal across all language generation tasks, not just math or code. To demonstrate this, we conducted an additional experiment on the **WritingBench** dataset. **Crucially, we utilized the exact same set of hyperparameters as in all other experiments in the paper.** The results are presented below:
> >
> > | Benchmark | Method | Length | Acc | Total Time (s) |
> > | :--- | :--- | :--- | :--- | :--- |
> > | **WritingBench** | Baseline | 64 | 16.314 | 1215 |
> > | | Baseline | 128 | 23.885 | 2271 |
> > | | Baseline | 256 | 31.608 | 5042 |
> > | | Baseline | 512 | 37.877 | 10616 |
> > | | Baseline | 1024 | 41.664 | 25404 |
> > | | **Baseline (Peak Acc)** | **2048** | **43.664** | **60599** |
> > | | **DAEDAL** | **Adaptive** | **43.245** | **11434** |
> >
> > As shown, even without any task-specific parameter tuning, DAEDAL achieves nearly identical peak performance (43.245 vs. 43.664) while reducing the total computation time from **60,599 seconds** to **11,434 seconds**. This represents a **~6$\times$ speedup**, further validating that DAEDAL successfully and efficiently generalizes to diverse, open-ended general language tasks.
> >
> > ---
> >
> > If you still have any concerns or aspects you would like to discuss further, please do not hesitate to contact us at any time.
> >
> > **We sincerely thank you for your time and thoughtful comments. If our response has addressed your concerns, we would deeply appreciate your consideration of raising your rating. We greatly value your feedback and, regardless, sincerely appreciate your engagement with our work.**
> >
> > Best regards,
> >
> > The Authors

---

### Official Review · Reviewer_owRq · 2025-10-31

**Soundness:** 3
**Presentation:** 3
**Contribution:** 2
**Rating:** 6
**Confidence:** 3

**Summary:**

This paper introduces an inference method for a pretrained masked diffusion model that supports length variability. In particular, the algorithm leverages two core operations-determining length using EOS confidence and then expanding iteratively for the masked position with the lowest prediction confidence. Experiments have been conducted on several math and code benchmarks on LLaDA-Instruct and Dream. The results show that the approach can maintain the accuracy of the "optimal" length sweeping over several lengths while being adaptive-length at inference time.

**Strengths:**

1. The goal of achieving adaptive-length decoding is promising and important to obtain better trade-off frontier between sample quality and latency.

2. The proposed approach is easy to follow and the method requires no additional training.

3. The presentation of the paper is clear and the empirical performance, specifically on the accuracy v.s. total tokens, is quite impressive. The ablation studies including hyperparameter sensitivity analysis are informative.

**Weaknesses:**

1. The proposed idea of using confidence of predicting EOS token to determine the length of the sequence seems a bit heuristic. A more comprehensive evaluation should be conducted on more datasets and tasks to justify the effectiveness of such heuristic.

2. Efficiency metrics such as wall-clock time/latency is missing.

**Questions:**

1. I am curious about the generalization of the findings in Figure 2 on over datasets and number of tokens (other than 128). Would the authors be able to provide more empirical justifications?

2. In the main table, it would be helpful to add actual wall-clock decoding time as a metric. Would the authors provide results for this?

3. Would the approach benefit from some training/finetuning of the model to better account for the variable-length objective?

---

> ### Author Response · Authors · 2025-11-21
>
> We sincerely thank Reviewer owRq for the positive assessment and for finding our approach **"easy to follow"**, the presentation **"clear"**, and the empirical performance **"quite impressive"**. We are also encouraged that Reviewer owRq **recognizes the importance of achieving adaptive-length decoding for obtaining a better trade-off frontier.**
>
> ---
>
> **Weakness 1: The proposed idea of using confidence of predicting EOS token to determine the length of the sequence seems a bit heuristic. A more comprehensive evaluation should be conducted on more datasets and tasks to justify the effectiveness of such heuristic.**
>
> **Question 1: I am curious about the generalization of the findings in Figure 2 on over datasets and number of tokens (other than 128). Would the authors be able to provide more empirical justifications?**
>
> **Response 1:** We appreciate the suggestion to validate the robustness of our core heuristic. In the revised paper, we have included **additional visualization results (Figure 6)** in appendix. These new results consistently align with our findings in Figure 2: the model's internal EOS confidence acts as a reliable signal for length sufficiency across various tasks and length settings. Specifically, we observe that when the allocated length is insufficient for the problem complexity, the EOS confidence at the sequence terminus remains consistently low, regardless of the specific dataset or token count, confirming that this "heuristic" effectively captures the model's intrinsic uncertainty about completeness.
>
> ---
>
> **Weakness 2: Efficiency metrics such as wall-clock time/latency is missing.**
>
> **Question 2: In the main table, it would be helpful to add actual wall-clock decoding time as a metric. Would the authors provide results for this?**
>
> **Response 2:** Thank you for this crucial suggestion. We have conducted wall-clock time measurements on **8 NVIDIA H800 GPUs**. We report the total computation time (batch size = 1) required for each fixed-length baseline configuration and for DAEDAL.
>
> * On GSM8K, to achieve a performance (83.8%) comparable to DAEDAL (85.8%), the baseline requires **11,545 seconds** of total computation. In contrast, DAEDAL achieves superior accuracy using only **1,049 seconds**, representing an **~11$\times$ speedup** in total computation time.
>
> * On MATH500, the baseline requires **14,196 seconds** for peak accuracy (39.6%), while DAEDAL achieves higher accuracy (44.2%) using only **1,160 seconds**, representing a **~12$\times$ speedup** in total computation time.
>
> | Benchmark | Method | Length | Acc | Avg Tokens | Total Time (s) |
> | :--- | :--- | :--- | :--- | :--- | :--- |
> | **GSM8K** | Baseline | 64 | 48.0 | 64 | 407 |
> | | Baseline | 128 | 67.9 | 128 | 837 |
> | | Baseline | 256 | 77.6 | 256 | 1825 |
> | | Baseline | 512 | 83.3 | 512 | 4405 |
> | | **Baseline** | **1024 (Peak Acc)** | **83.8** | 1024 | 11545 |
> | | Baseline | 2048 | 82.6 | 2048 | 37016 |
> | | **DAEDAL** | **Adaptive** | **85.8** | 363 | **1049** |
> | **MATH500** | Baseline | 64 | 24.0 | 64 | 177 |
> | | Baseline | 128 | 29.0 | 128 | 340 |
> | | Baseline | 256 | 35.6 | 256 | 717 |
> | | Baseline | 512 | 38.8 | 512 | 1690 |
> | | Baseline | 1024 | 39.4 | 1024 | 4444 |
> | | **Baseline** | **2048 (Peak Acc)** | **39.6** | 2048 | 14196 |
> | | **DAEDAL** | **Adaptive** | **44.2** | 704 | **1160** |
>
> ---
>
> **Question 3: Would the approach benefit from some training/finetuning of the model to better account for the variable-length objective?**
>
> **Response 3:** This is an excellent point for future exploration. While DAEDAL is designed as a training free method to unlock the latent capabilities of existing pre-trained DLLMs, we agree that fine tuning the model specifically with variable-length objectives could likely further enhance its sensitivity to length requirements and potentially improve the planning capability.
>
> ---
>
> If you still have any concerns or aspects you would like to discuss further, please do not hesitate to contact us at any time.
>
> **We sincerely thank you for your time and thoughtful comments. If our response has addressed your concerns, we would deeply appreciate your consideration of raising your rating. We greatly value your feedback and, regardless, sincerely appreciate your engagement with our work.**
>
> Best regards,
>
> The Authors

---

### Official Review · Reviewer_xU9B · 2025-11-02

**Soundness:** 2
**Presentation:** 3
**Contribution:** 2
**Rating:** 6
**Confidence:** 3

**Summary:**

This paper proposes an inference-time method for extending the generation length of text diffusion models. Starting with a small canvas, masked tokens get added until EOS occurs with high-enough probability at the end of the sequence. Given this initial sequence of masked tokens, the method alternates between filling in high-confidence tokens and adding more tokens based on EOS confidence. Tokens are added to the lowest confidence mask positions, and are expanded by a constant block size.

Results on simple math and coding evaluations show that the method obtains reasonable accuracy while adaptively determining response length. The average ratio between utilized tokens versus padding is around 65% across all tasks.

**Strengths:**

The method seems effective and is very simple. The presentation was written to be reader-friendly, but I believe would benefit from including more details. For example, Algorithm 1 has a few more details than Fig 3. The results are reasonable, with easing the need for manual sequence length tuning.

**Weaknesses:**

The main weakness is that the main baselines to compare to is block diffusion or other adaptive-length methods. The sell of variable-length  diffusion would likely have to be a speed increase while preserving accuracy, or another point on the speed-accuracy Pareto frontier. It is possible that spec-decoded autoregressive models achieve better speeds at similar accuracy.

**Questions:**

The paper supports its claims. The main improvement would be to include speed-accuracy comparisons to other accelerated methods, namely speculative decoding.

---

> ### Author Response · Authors · 2025-11-21
>
> We sincerely thank Reviewer xU9B for the constructive feedback. We are particularly encouraged that you recognized our method as **effective and very simple**, the paper as **reader friendly**, and most importantly that DAEDAL successfully **eases the need for manual sequence length tuning**.
>
> ---
>
> **Weakness 1: The main weakness is that the main baselines to compare to is block diffusion or other adaptive-length methods. The sell of variable-length diffusion would likely have to be a speed increase while preserving accuracy, or another point on the speed-accuracy Pareto frontier. It is possible that spec-decoded autoregressive models achieve better speeds at similar accuracy.**
>
> **Response 1:** We appreciate the reviewer's insight regarding the value proposition of variable length diffusion. While we agree that efficiency is a major benefit, we emphasize that the primary "sell" of DAEDAL is **resolving the architectural constraint** of native DLLMs. It allows the model to break free from the **Static Length Dilemma**. As shown in our experiments (Figure 1), a fixed length that is too short leads to failure, while a length that is universally too long not only incurs massive quadratic overhead but can also **result in performance degradation**. DAEDAL solves this by adaptively finding the optimal length per problem, avoiding both under generation and the negative impact of over generation. Regarding the comparison with Speculative Decoded Autoregressive models, we respectfully argue that comparing directly with them would conflate different generation paradigms. Our work focuses on optimizing the **native** Diffusion LLM framework which utilizes **global planning** , whereas AR models operate sequentially. Therefore, the most meaningful evaluation is demonstrating how DAEDAL optimizes correctness and resource usage within the diffusion paradigm itself, rather than racing against the AR paradigm.
>
> ---
>
> **Question 1: The paper supports its claims. The main improvement would be to include speed-accuracy comparisons to other accelerated methods, namely speculative decoding.**
>
> **Response 2:** Thank you for the suggestion to compare with accelerated methods like Speculative Decoding. **Speculative Decoding is based on the autoregressive paradigm and is not a feature implemented in Diffusion LLMs. Conversely, our DAEDAL method relies on the bidirectional masking and denoising mechanism inherent to DLLMs and cannot be applied to AR LLMs for a fair comparison of the variable length mechanism.** To address your concern, we provide a **theoretical efficiency analysis**. As shown in **Table 1** for GSM8K, the baseline requires a fixed computation of **1024 tokens** to achieve peak performance (Acc 83.8%). In stark contrast, DAEDAL achieves superior accuracy (**85.8%**) using an average of only **363 tokens**. This represents a **2.8 times reduction** in computational load. Simultaneously, the **Effective Token Ratio** dramatically increases from 27.7% to 73.5%. These results demonstrate that DAEDAL is highly effective at optimizing efficiency within the DLLM framework. However, given that our contribution is specific to the Diffusion LLMs, we believe it would be methodologically incongruous to benchmark this mechanism against the fundamentally different Speculative Decoded AR LLMs.
>
> ---
>
> If you still have any concerns or aspects you would like to discuss further, please do not hesitate to contact us at any time.
>
> **We sincerely thank you for your time and thoughtful comments. If our response has addressed your concerns, we would deeply appreciate your consideration of raising your rating. We greatly value your feedback and, regardless, sincerely appreciate your engagement with our work.**
>
> Best regards,
>
> The Authors

---

### Meta-Review · Area_Chair_Frw5 · 2026-01-07

**Summary:**

The reviewers found the paper’s idea of training-free variable-length decoding for diffusion LLMs to be simple, well-motivated, and empirically promising, but initially raised concerns about missing efficiency metrics, robustness of the EOS-confidence heuristic, generalization beyond math/code tasks, and clarity on the necessity of the two-stage design. The authors’ rebuttal adds substantial new experimental evidence and clarifications that directly address these concerns, strengthening confidence in both the empirical results and the overall contribution.

**Reviewer Concerns:**

In the rebuttal, the authors addressed concerns by adding extensive latency measurements showing substantial (6–12×) reductions in total computation time at equal or better accuracy, providing additional analyses demonstrating that the confidence signal behaves consistently across datasets, lengths, and models using a single set of hyperparameters, and extending the evaluation to a general-language benchmark with similar conclusions. The authors also clarified the complementary roles of the two stages—global planning and local refinement—and supported this explanation with ablation results. While comparisons remain limited to the diffusion paradigm, this is a reasonable methodological choice given the scope of the paper, and no major concerns remain outstanding after the rebuttal.

**Reviewer Scores:**

All three reviewers initially gave scores of 6, indicating marginally above-threshold acceptance with some reservations. Given the additional empirical evidence and clarifications provided in the rebuttal, it is likely that reviewers would have increased confidence in their assessments.

---

### Decision · Program_Chairs · 2026-01-26

Accept (Poster)